# Electrochemical Characterization of Nanoporous Alumina-Based Membranes with Different Structure and Geometrical Parameters by Membrane Potential Analysis

**Virginia Romero and Juana Benavente ***

Departamento de Física Aplicada I, Facultad de Ciencias, Universidad de Málaga, E-29071 Málaga, Spain; virgirom@uma.es
* Correspondence: j_benavente@uma.es

**Abstract:** Electrochemical characterization of alumina-based membranes obtained by two different techniques, sinterization or anodization, is performed by analyzing membrane potential values. This analysis allows us the estimation of the effective concentration of fixed charge in the membrane ($X_{ef}$) and the transport number of the ions into the pores ($t_i$), as well as the determination of ionic permselectivity ($P(i)$) and their correlation with the different structures (supported, symmetrical or asymmetric), geometrical parameters (pore size and porosity) and surface materials (alumina-zirconia or alumina) of the studied membranes. From these results, the electropositive character of the membranes was stated, but also the significant reduction (70%) in $X_{ef}$ value and around 30% in permselectivity when pore size increases from 25 nm to 100 nm, in the case of sinterized alumina-zirconia membranes with similar porosity (CRF samples), while the clear influence of pore size on the electrochemical behavior of the electrochemically synthesized alumina membranes (NPAM samples) was confirmed as well as the lower influence of membrane porosity. Moreover, the effect of protein (BSA) static fouling on electrochemical parameters for both CRF and NPAMs samples was also analyzed, and our results show a reduction in the electropositive character of both membranes, being this behavior opposite to that discussed for one of the NPAMs as a result of surface modification with a theophylline derivative (Theo **1**).

**Keywords:** nanoporous membranes; membrane potential; effective fixed charge concentration; permselectivity

## 1. Introduction

Although the use of inorganic membranes started at the end of the 1940s, with application in uranium concentration and gas separation due to their chemical and thermal resistance, its development has increased more recently due to their application in separation processes related to food, biochemical and pharmaceutical industries but also for oily waters and wastewaters treatments [1–5], which usually work with solutions containing different kinds of macromolecules (proteins, colloids, etc.) able to cause membrane fouling, that is, the deposition/adsorption of particles (neutral or charged) from solutions on the membrane surface and pore walls, since they support cleaning protocols (hard chemicals, high temperature or even radiation) better than polymeric membranes. In fact, membrane fouling, which reduces the permeate flux but also affects the separation properties of a membrane, is the main factor limiting the use of membranes in the filtration process [6]. Consequently, the high thermal and chemical resistance of inorganic membranes are properties very helpful for those applications due to the use of different chemicals and anti-scaling agents. Additionally, the high tensile modulus of inorganic materials favors the resistance of inorganic membranes to compaction under high pressure (mechanical stability), and other factors such as low sensitivity to bacterial action and long operational life also play in favor of them when compared with polymeric ones [5], but their higher cost, when

compared with polymeric membranes, reduces their use in many common applications. In the case of filtration processes, inorganic membranes use to have a composite structure consisting of porous support covered by one or more layers of the same or different materials (alumina, silica, zirconia, silicon carbide, . . . ). Depending on the size of the layered materials, microfiltration, ultrafiltration, or nanofiltration membranes can be obtained.

Among the different techniques used for the fabrication of membranes, sintering is a simple process that allows obtaining porous membranes by compressing powders composed of particles of known sizes at high temperature, being the structural characteristic parameters of membranes (pore size, porosity, and tortuosity) directly related with sintering temperature as well as particle size and pressure, depending the required temperature on the material used. Due to the fabrication process, these membranes do not exhibit a unique pore size value but a certain range of values, and tortuosity [1], differing significantly from the structure associated with the model (or ideal) porous membranes.

On the other hand, fabrication by an electrochemical process of nanoporous alumina structures (NPASs) with self-ordered growth was reported in 1995 [7]. These NPASs were obtained by electrochemical anodization of aluminum foils (two-step anodization method), and they were initially used as templates for nanotubes, nanodots, or nanowires [8,9]. The NPASs present a series of parallel and cylindrical capillaries with pore sizes ranging between 20–200 nm and porosity between 10–30%, depending on anodization conditions (mainly anodization potential, concentration/type of electrolyte solution, and temperature), with narrow pore size distribution and without tortuosity, and thickness between 10 μm and 100 μm [10]. Due to their well-defined structure, similar to an "ideal" nanoporous membrane, these alumina nanoporous structures were also used as membranes for the separation of radioactive cations, drug delivery, nanofluids, medical devices, or even photonic crystals [11–15]. Although electrochemical conditions control the pore-size/porosity of these nanoporous alumina membranes (NPAMs), chemical etching allows a slight increase in pore size and porosity, while a reduction in both geometrical parameters can also be obtained by atomic layer deposition (ALD) technique [16,17], which permits us to obtain a set of NPAMs with symmetrical structure and covering a rather wide interval of pore-size/porosity values, although asymmetric membranes can also be obtained by sequential modification of electrochemical conditions [18]. Moreover, the use of different ceramic oxides ($SiO_2$, $TiO_2$, $Fe_2O_3$, . . . ) as membrane surface coating layer (both external and pore-wall surfaces) in the ALD process (monolayer or multilayer process) also allows the control of other surface characteristics such as hydrophobicity/hydrophilicity, electrical character or even optical properties, depending on the material selected for a specific application, opening the application of these nanoporous alumina-based structures or membranes [19–23].

As it is well known, membranes used in separation processes can differ significantly in material and structure, being a necessary adequate characterization to select the membrane most convenient for a given process. Membrane material, structure (porous or dense, symmetric or asymmetric, uniform or composite), and electrical (positively/negatively charged) character are basic information needed previous to the election of a membrane for a particular separation process, which can be obtained by very different characterization techniques. In the case of electrical characterization, three different techniques, membrane potential, streaming potential, and impedance spectroscopy measurements, are commonly used. Conductivity and dielectric constant of homogeneous membranes (but also the dense and porous sublayers of reverse osmosis and nanofiltration membranes) are determined by impedance spectroscopy measurements using equivalent circuits as models [24–26], although it can also be used for estimation of changes associated to membrane fouling or surface modification (interfacial effects) [27–29]; membrane potential and streaming potential measurements give information on the electropositive/electronegative character of membranes, and they allow the estimation of different electrochemical parameters as well as the effect of different kinds of membrane modification (fouling, material or surface modification) [30–32].

In this work, electrochemical characterization of nanoporous alumina-based membranes with different structures (composite or monolayer), geometry (pore size and porosity), fabrication method (sintered/electrochemical), and applications (filtration or controlled diffusion devices, mainly) was performed by analyzing membrane potentials (electrical potential difference at both sides of a membrane separating two electrolyte solutions of different concentrations) using the Teorell–Mayer–Sievers (TMS) model [33,34], which also allows a separate estimation of diffusive and interfacial contributions. From membrane potential results, the electropositive character of the different membranes studied as well as the effect of pore size, porosity and/or membrane structure on the effective concentration of fixed charge in the membrane ($X_{ef}$), the cation/anion transport numbers ($t_i$) through the membrane pores and its ionic permselectivity ($P(i)$) were determined. In particular, the comparison of the electrochemical parameter determined for the studied membranes shows the significant reduction in both membranes' fixed charge concentration and anionic permselectivity with the increase in pore size, independently of membrane structure (symmetric/asymmetric; ideal straight uniform pores or tortuosity and pore size distribution) and surface material, with only small effects of the other geometrical characteristics such as porosity; the influence of another experimental factor such as concentration polarization is also indicated. Moreover, membrane potential analysis also allows us the estimation of changes in the membrane electrochemical parameters as a result of immobilization of an active pharmaceutical substance (Theo **1**) in its structure [35] or protein (BSA) fouling by comparing the values obtained for clean and modified samples, which can be taken as an indirect way of membranes modification confirmation and/or estimation of fouling mechanisms.

## 2. Materials and Methods

### 2.1. Material

Two planar and flexible commercial composite alumina-zirconia membranes for filtration application (CREAFILTER, Degussa, Germany) obtained by sintering process [36], which consist of a fibrous stain steel network covered by a sublayer of $Al_2O_3$ particles plus an external layer of $ZrO_2$ particles (5 μm thickness and 45% porosity), with total thickness of around 80 μm (according to supplier). Membrane fabrication process consists of different steps [36] and, in particular, after first heating at 250 °C, the temperature is elevated up to 650 °C for membrane stabilization, although they remain flexible. These membranes exhibit different average nominal pore sizes, 25 nm or 100 nm, and they will hereafter denominate CRF25 and CRF100, respectively. SEM pictures of surface and cross-section of the CRF100 membrane are presented as Supplementary Information (Figure S1).

Two nanoporous alumina membranes (NPAMs) obtained by the two-step anodization process, with a thickness of around 60 μm, one commercial (And-NPAM) and the other experimental (Sfw-NPAM) were also studied. The commercial And-NPAM sample (Anodisc[TM] from Whatmann, Maidstone, UK) has asymmetric structure, with an average pore size of 20 nm for one surface but 200 nm for the other surface (supplier information) and porosity ranging between 25–35%. The Sfw-NPAM sample has symmetric structure with an average pore size of 28 nm and porosity of 17%. This sample was synthesized by Prof. V. de la Prida and Dr. V. Vega (Applied Physic Department and Nanomembranes Laboratory, University of Oviedo, Oviedo, Spain) using an aluminum disc of high purity (Al 99.999%, with 25 mm diameter of 25 mm and 0.5 mm of thickness) and 0.3 M solution of sulfuric acid at a constant anodization voltage of 25 V, and pore opening by immersion in a phosphoric acid solution (5%) for 8 min; it should be indicated that different chemicals (isopropanol, ethanol, perchloric acid, ethanol, $CrO_3$, and $H_3PO_4$ solutions or aqueous mixtures of HCl and $CuCl_2$ or $H_3PO_4$ 5%) were also used during the different steps of membrane synthesis (initial cleaning and final pores bottom opening [37]). Surface SEM pictures of Sfw-NPAM and And-NPAM samples, as well as partial cross-section of this latter membrane showing pore size change, are also presented as Supplementary Information in Figure S1.

Moreover, possible electrochemical changes associated with protein fouling of the CRF25 and the And-NPAM samples were also considered. Both membranes were statically fouled with an aqueous solution containing 5 g/L of bovine serum albumin (BSA, 66.5 kDa molecular weight, Stokes radius of 3.48 nm, and isoelectric point ~4.8) by contacting one of the membrane surfaces with the BSA solution and the other surface with distilled water for 24 h, being these sample called CRF25(f) and And/BSA-NPAM, respectively. Additionally, results previously obtained for the And-NPAM sample after inclusion of a pharmacologically active molecule, the theophylline-oligo(ethylene glycol)-alkene derivative **1** (or Theo **1**), are also presented for comparison reasons (And/Theo **1**-NPAM sample) [35]; a scheme of Theo **1** as well as indication on sample preparation is given as Supplementary Information (Figure S2).

*2.2. Membrane Potential Measurements*

Membrane potential, or the equilibrium electrical potential difference between two electrolyte solutions of different concentrations (Cf and Cv) separated by a membrane, was measured in a dead-end test cell consisting of two glass half-cells [38], with a magnetic stirrer in the bottom of each half-cell (stirring rate of 550 rpm) to minimize the concentration-polarization at the membrane surfaces [38]. These measurements were carried out with NaCl solutions (at 25 ± 2 °C and pH = 5.9 ± 0.2) by keeping fixed the concentration of the solution at one side of the membrane (Cf = 0.01 M) and gradually changing the concentration of the solution at the other side ($0.002 M \leq C_v \leq 0.1 M$); moreover, measurements at a fixed concentration $C_f = 0.001 M$ were also performed with some membranes and, after each series of measurements, the membranes were kept in contact with distilled water. A Ag/AgCl electrode (reversible to ion $Cl^-$) was placed in each half-cell and connected to a digital voltmeter (Yokohama 7552, 1GΩ input resistance) for cell potential (ΔE) measurements. Membrane potential ($\Delta\Phi_{mbr}$) values were obtained by subtracting the electrode potential to cell potential values for each pair of $C_v/C_f$ concentrations, that is, $\Delta\Phi_{mbr} = \Delta E - \Delta\Phi_{elect}$, being $\Delta E = -(RT/Fz)\ln(C_v/C_f)$, where R and F represent the gas and Faraday constants, z is the ion valence and T the temperature of the system.

## 3. Results and Discussion

Electrochemical characterization of the studied membranes was carried out by analyzing membrane potential values, which allow the determination of the effective concentration of fixed charge in the membrane, $X_{ef}$, and the transport number of the ions (cation or anion) from solutions through the membrane pores, ti, which represent the fraction of the total current transported for each ion, that is: ti = Ii/IT [39]. For positively charged membranes, the transport number of cations ($t_+$), or membrane co-ions, is lower than in solution ($t_+{}^o$) due to electrostatic repulsion, with $t_+ = 0$ (and consequently, $t_- = 1$) in the extreme case of ideal anion exchanger membranes, and the opposite ($t_+ = 1$ and $t_- = 0$) would correspond to ideal cation exchange membranes (negatively charged membranes). $X_{ef}$ and $t_i$ values can be determined from membrane potential data points ($\Delta\Phi_{mbr}$) by means of the Teorell–Meyer–Sievers (or TMS) model [33,34], which assumes, for charged membranes, three contributions for membrane potential: (i) a Donnan potential ($\Delta\emptyset Don(i)$) at each membrane-solution interface due to the (partial) electrical exclusion of co-ions; (ii) a diffusion potential in the membrane caused by the different mobility of the ions inside the pores; that is: $\Delta\Phi_{mbr} = \Delta\emptyset Don(I) + \Delta\emptyset dif + \Delta\emptyset Don(II)$, where (I) and (II) indicate each membrane/solution interface. Taking into account the expressions for Donnan and diffusion potentials (for 1:1 electrolytes) [39]:

$$\Delta\emptyset Don(j) = (RT/F) \cdot \ln[(w \cdot Xef/2Cj) + [(w \cdot Xef/2Cj)^2 + 1]^{1/2}]] \quad (1)$$

$$\Delta\emptyset dif = (RT/F)[(t_- - t_+)] \cdot \ln(C_f/Cc_v) = (RT/F)[(1 - 2t_+)] \cdot \ln(C_f/C_v) \quad (2)$$

then, the membrane potential would be expressed as follows [39]:

$$\Delta\Phi_{mbr} = -\frac{RT}{wzF}\left[U\ln\frac{\sqrt{4y_v^2+1}+wU}{\sqrt{4y_f^2+1}+wU} - \ln\frac{cf}{cv}\frac{\sqrt{4y_v^2+1}+w}{\sqrt{4y_f^2+1}+w}\right] \tag{3}$$

where w = +1/−1 for positively/negatively charged membranes, $U$ is a parameter related to the diffusion of ions in the membrane pores ($U = ((D_+ - D_-)/(D_+ + D_-) = (t_+ - t_-) = (2t_+ - 1)$ for 1:1 electrolytes), while $y_i = C_i/|X_{ef}|$ (I = $f$ or $v$), since the same value of concentration in the solution just outside/inside the pore is assumed for porous membranes (partition coefficient value = 1) [37], and the other parameters have already been indicated. Previously to analyze the studied membranes, in Supplementary Information (Figure S3), an example of the capability of this analysis to discriminate different membrane factors such as material (alumina or regenerated cellulose) and geometrical characteristics (pore size and porosity or membrane swelling, respectively) is presented. As it can be observed, independently of membrane material and geometrical parameters, at low concentrations (C << C$_f$), $\Delta\Phi_{meb}$ values are more similar to that exhibited by ideal ion-exchange membranes due to the exclusion of solution co-ions by electrical interactions with membranes fixed charges, being the transport basically associated to counter-ions; however, at high solution concentrations (C >> C$_f$), when interfacial effects are partially/totally masked by the free solution charges (ions), Donnan contribution is clearly reduced, except in the case of ion-exchange membranes (see Figure S3c).

Variation of membrane potentials with the ratio of solution concentrations at both sides of the CRF25 and CRF100 membranes is shown in Figure 1a and, for comparison reasons, theoretical values for an ideal positively/negatively charged membrane ($t_+$ = 0/1, respectively) and the NaCl solution diffusion potential ($\Delta\Phi_{dif}^o$) due to the different values of ions transport numbers in solution (characteristic of each electrolyte solution) are also indicated in Figure 1a by solid lines and dashed line, respectively; Figure 1b shows a scheme of pore size effect on ion's transport for a model or ideal positive membrane with the same value of fixed charge, while the effect of different fixed charge value on the transport of ions through two ideal membranes with similar pore size is shown in Figure 1c. It should be pointed out that fixed charges located on particular points of non-ideal porous membranes, such as those obtained sintered process, might significantly affect the transport of ions.

The first qualitative information from $\Delta\Phi_{mbr}$ values shown in Figure 1a is the electropositive character of both sintered membranes (a similar tendency to the anion exchanger) and the same kind of $\Delta\Phi_{mbr} - \ln(C_v/C_f)$ dependence as that shown in Figure S3b for almost ideal alumina nanoporous membranes with different pore size but similar porosity. In fact, only slight differences in $\Delta\Phi_{mbr}$ values for a given pair of NaCl solution concentrations can also be observed when results obtained for both membranes are compared at both low and high solution concentrations, although in opposite ways, which might be related to high interfacial contribution at low concentrations (C$_v$ < C$_f$) for the CRF25 membrane, or higher contribution of ions diffusion through the pores at high concentration (C$_v$ > C$_f$) in the case of the CRF100 membrane, as it was already reported for symmetric unsupported nanoporous alumina membranes [37].

The fit of the $\Delta\Phi$mbr data to Equation (3) allows us the estimation of $X_{ef}$ and $t_i$ values, and the results obtained for both membranes (as well as the fit error) are indicated in Table 1. Figure 2 shows a comparison of experimental (points) and theoretical (solid line) values for CRF25 and CRF100 membranes, plus the dependence with a concentration ratio of both Donnan potential (dashed-dot line) and diffusion potential through the membrane pores (dashed line) contributions; these results show the dominancy at high C$_v$ values of diffusion potential contribution when compared with the Donnan potential contribution, in agreement with the rather low values of membranes fixed charge concentration (low interfacial effect). On the other hand, it could also be of interest to compare $t_+$ values shown in Table 1 with those previously reported for both membranes (<$t_+$$^{ap}$>$^{CRF25}$ = 0.340 ± 0.003 and <$t_+$$^{ap}$>$^{CRF100}$ = 0.341 ± 0.004 [40]), which were determined without taking into account

the effect of membrane fixed charge, that is, assuming that membrane potential values correspond only to the solution diffusion in the membrane pores, without considering the interfacial or Donnan effect (apparent ionic transport numbers or $t_i^{ap}$). No differences between $t_+$ and $t_+^{ap}$ for the CRF100 membrane (0.5%) exist due to its relatively large pore size, but in the case of the CRF25 membrane, such a difference is 9%.

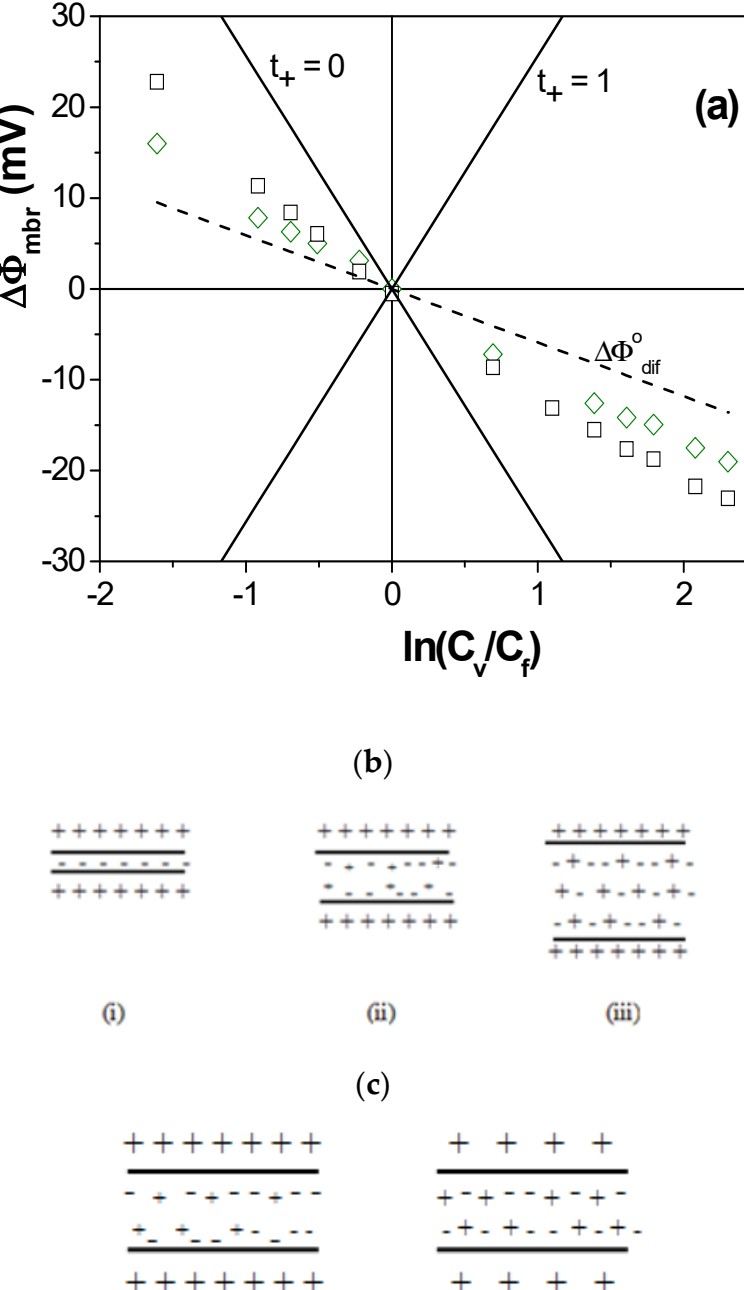

**Figure 1.** (**a**) Membrane potential dependence with solution concentrations ratio for CRF25 (□) and CRF100 (◊) membranes; NaCl solution diffusion potential (dashed line), ideal anion exchange membrane (solid line, $t_+ = 0$) and ideal cation exchange membrane (solid line, $t_+ = 1$). (**b**) (**i**) Ideal exchanger ($t_+ = 0$); (**ii**) Pore partial control; (**iii**) No pore control. Scheme of pore size effect on ion transport through an ideal nanoporous positively charged membranes and (**c**) effect of (positive) fixed charge on ion transport through ideal nanoporous membranes with similar pore size. Non-ideal porous structure might differently affect ions transport.

**Table 1.** Fixed charge concentration in the membrane ($X_{ef}$), cation ($t_+$) and anion ($t_-$) transport numbers in the membrane, fit error and anionic permselectivity (P(-)) values for membranes CRF25, CRF100 and CRF25(f).

| Membrane | $X_{ef}$ (M) | $t_+$ | $t_-$ | Fit Error | P(-)% |
|----------|--------------|-------|-------|-----------|-------|
| CRF25 | $+4.4 \times 10^{-3}$ | 0.312 | 0.688 | 7.4% | 18.9 |
| CRF100 | $+1.3 \times 10^{-3}$ | 0.338 | 0.662 | 8.4% | 12.2 |
| CRF25(f) | $+3.0 \times 10^{-3}$ | 0.328 | 0.672 | 8.6% | 14.8 |

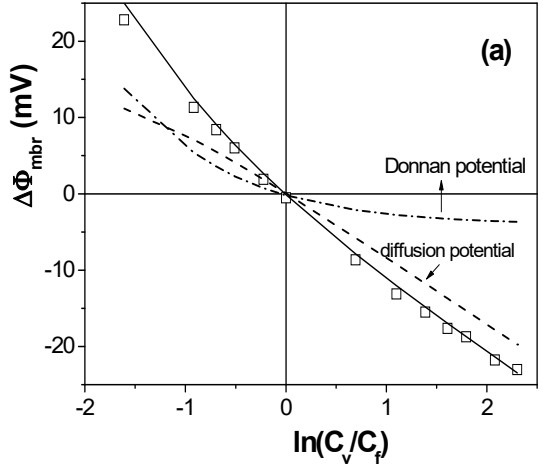
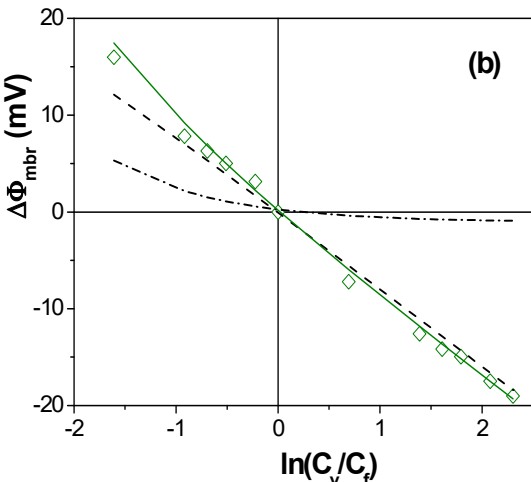

**Figure 2.** Comparison of experimental (points) and fitted (solid line) values, Donnan (dashed dot line) and diffusion (dashed line) contributions. (**a**) CRF25 (□); (**b**) CRF100 (◇).

Ionic (anionic or cationic) permselectivity (P(i)) is another electrochemical parameter of interest since it gives information on the preferential permeation of an ionic specie through a charged membrane, being a measure of the selectivity of a membrane toward the passage of counter-ions; anion permselectivity can be determined by the following expression [39]: P(-) = $(t_- - t_-^o)/t_+^o$, where $t_-$ indicates the anion transport number in the membrane, while $t_-^o$ and $t_+^o$ represent the anion/cation transport numbers in solution (average values for NaCl solutions: $<t^o_{Cl-}> = 0.615$ and $<t^o_{Na+}> = 0.385$ [41]). P(-) values for CRF25 and CRF100 membranes are also indicated in Table 1. These results show a decrease of 35% in the anionic permselectivity with an increase of four times in pore radii.

On the other hand, it should be indicated that for a given membrane and electrolyte solution, concentration level and stirring solutions can also affect membrane potential values, as can be observed in Figure 3, where variation of $\Delta\Phi_{mbr}$ values with the average concentration of the solutions at both membrane sides ($C_{avg} = (C_f + C_v)/2$) for membrane CRF25 at two different $C_f$ values ($C_f$ = 0.01 M or 0.001 M NaCl) is indicated, which is associated to the effect of concentration polarization at the solution/membrane interfaces [38].

Electrochemical modification of a membrane caused by protein fouling can also be determined by comparing membrane potential values for clean and fouled samples. A comparison of membrane potential values obtained for CRF25 and CRF25(f) membranes is shown in Figure 4a, where differences due to the effect of BSA fouling can be observed. In fact, $\Delta\Phi_{mbr}$ values for the CRF25(f) membrane are slightly lower than those determined for the clean one, for both low and high concentrations, which seems to indicate a slight reduction of the electropositive character of the CRF25 sample associated with the electronegative character of BSA at solutions pH (BSA isoelectric point ~4.8), indicating the presence of BSA on the membrane surface but also on the pore walls. Values determined for membrane fixed charge concentration, cation, and anion transport numbers in the CRF(f) membrane, as well as anionic permselectivity, are indicated in Table 1, while Figure 4b shows the comparison between experimental (points) and theoretical (solid line) membrane

potential values as well as the Donnan potential (dashed-dot line) and diffusion potential (dashed line) contributions for the CRF25(f) membrane. A comparison of the electrochemical parameters determined for clean and fouled membranes indicates a reduction of around 30% for the effective fixed charge concentration and 20% for the anionic permselectivity as a result of BSA fouling.

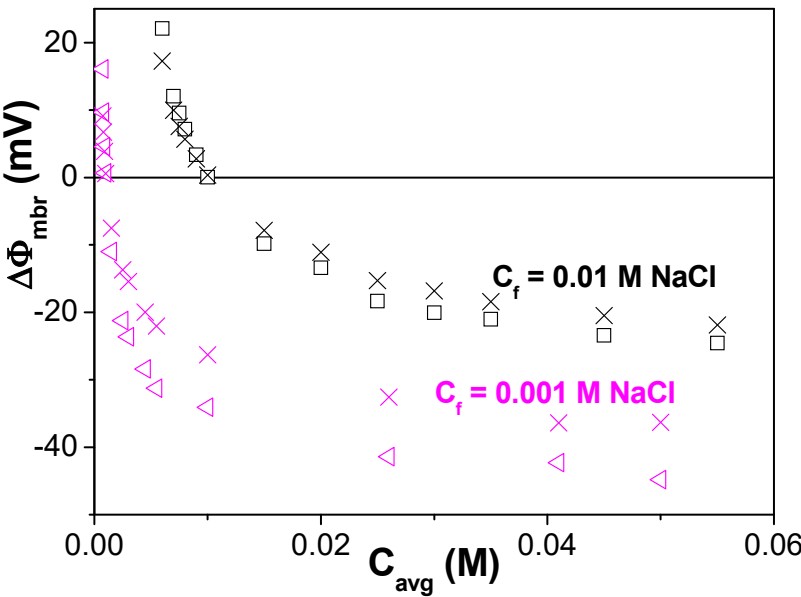

**Figure 3.** Variation of membrane potential with average concentration of the solution at both membranes sides the CRF25 membrane: (□, ◁) stirred solutions, (×) non stirred solutions. $C_f$ = 0.01 M NaCl (black points), $C_f$ = 0.001 M NaCl (magenta points).

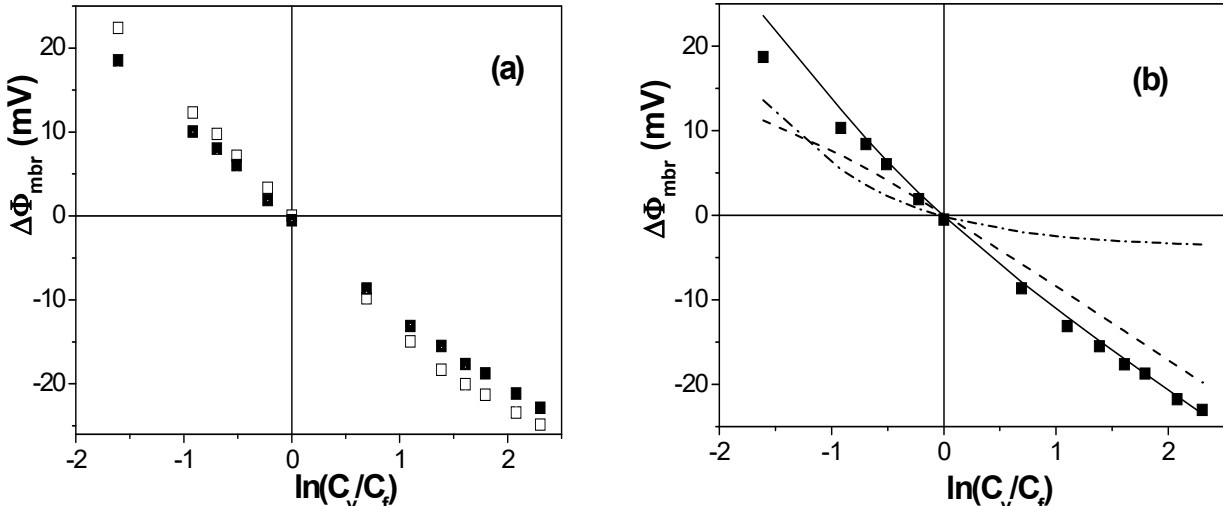

**Figure 4.** (**a**) Comparison of membrane potential dependence with solution concentrations ratio for CRF25 (□) and CRF25(f) (■) membranes; (**b**) comparison of experimental (■) and fitted (solid line) values for the CRF25(f) membrane, Donnan (dashed dot line) and diffusion (dashed line) contributions.

Electrochemical characterization of the two NPAMs by membrane potential analysis was also performed, and Figure 5a shows a comparison of $\Delta\Phi_{mbr}$ values dependence with concentrations ratio for Sfw-NPAM and And-NPAM samples; the theoretical values for an ideal positively charged membrane and the NaCl solution diffusion potential ($\Delta\Phi_{dif}{}^o$) are also indicated. As it can be observed, there are significant differences in $\Delta\Phi_{mbr}$ values obtained for both membranes; in particular, the values corresponding to the Sfw-NPAM

sample are rather similar, at low solution concentrations, to that presented by an ideal anion-exchanger, although the difference is higher at high concentrations (~25% for $C_v > 0.04$ M), while $\Delta\Phi_{mbr}$ values for the And-NPAM sample are very similar to the solution diffusion potential for the whole interval of concentrations, indicating low membrane control on the transport of ions. In fact, these results are rather similar to those already shown in Supplementary Information (Figure S3b) for NPAMs with lower/larger pore sizes, although in such cases, the membranes had practically the same porosity. Unfortunately, due to the different structures of both membranes (symmetric with an average pore size of 28 nm the Sfw-NPAM, but asymmetric with 20 nm/200 nm pore size the And-NPAM) and even porosity (15% or 25–35%, respectively), it is not possible *a priori* to ascribe changes in membrane potential values to a particular geometrical difference; consequently, for clarification reason, $\Delta\Phi_{mbr}$ values for the And-NPAM, the CRF100 membrane previously studied (100 nm pore size and 45% porosity) and those already obtained for a symmetric NPAM with 180 nm average pore size and 10% porosity (sample Ph-NPAM, [42]) are compared in Figure 5b. These results show, qualitatively, the reduced influence of porosity, or even membrane material (alumina or alumina-zirconia), on ionic diffusive transport when pore size values are around 100 nm (or higher), although the difference in surface material may be responsible for the differences in $\Delta\Phi_{mbr}$ values observed in Figure 5b at the lowest concentrations. The fit of the experimental data to Equation (3) allows us to estimate $X_{ef}$, ti, and P(-) for And-NPAM and Sfw-NPAM samples, and their values are indicated in Table 2. These results show a reduction higher than 75% in the value of the effective fixed charge concentration and 60% in anionic permselectivity with pore size increase higher than four times, independently of their symmetric or asymmetric structure. On the other hand, a comparison of values indicated in Tables 1 and 2 for CRF100 and And-NPAM confirms the quantitative similarity of the electrochemical behavior of membranes with pore size around 100 nm (or higher), while in the case of CRF25 and Sfw-NPAM samples, with 25 nm or 28 nm pore size, the higher porosity exhibited by the CRF25 membrane (~2.5 higher than the Sfw-NPAM) might be mainly responsible for 30% reduction in permselectivity when compared with the NPAM one.

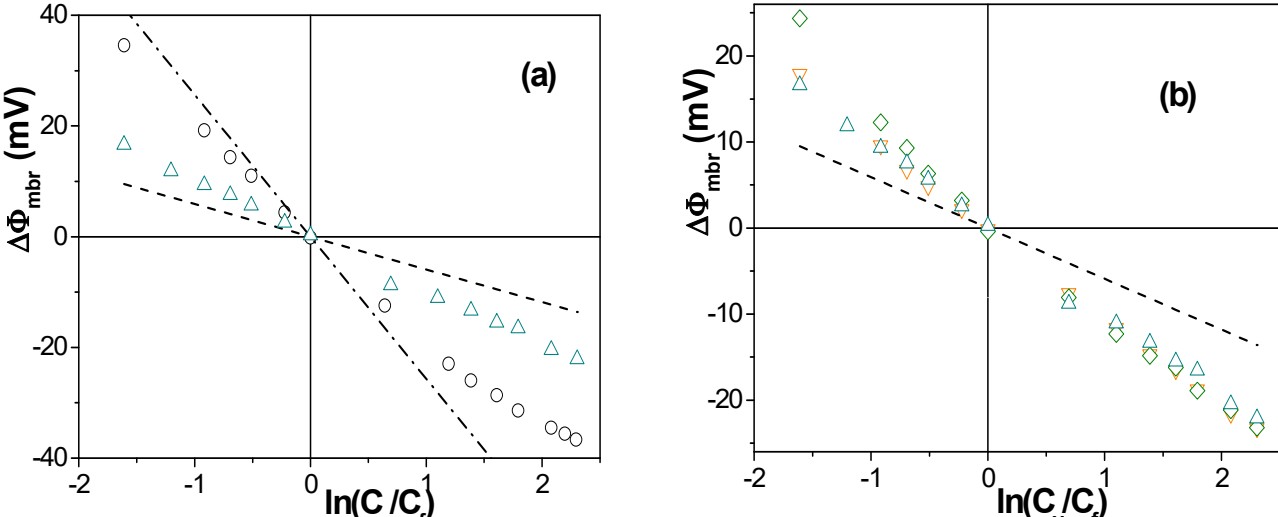

**Figure 5.** Variation of membrane potential values with solution concentration ratio: (**a**) Sfw-NPAM sample (o), And-NPAM sample ($\triangle$); (**b**) CRF100 ($\diamond$), Ph-NPAM ($\triangledown$), And-NPAM ($\triangle$). NaCl solution diffusion potential (dashed line) and ideal ion exchange membrane potential (dashed-dot line).

**Table 2.** Fixed charge concentration in the membrane ($X_{ef}$), cation ($t_+$) and anion ($t_-$) transport numbers in the membrane, fit error and anionic permselectivity (P(-)) values for membranes Sfw-NPAM, And-NPAM, And/BSA-NPAM and And/Theo1-NPAM samples.

| Membrane | $X_{ef}$ (M) | $t_+$ | $t_-$ | Fit Error | P(-)% |
|---|---|---|---|---|---|
| Sfw-NPAM | $+6.5 \times 10^{-3}$ | 0.287 | 0.713 | 3.1% | 25.5 |
| And-NPAM | $+1.3 \times 10^{-3}$ | 0.347 | 0.653 | 5.8% | 9.9 |
| And/BSA-NPAM | $+0.3 \times 10^{-3}$ | 0.338 | 0.662 | 6.8% | 12.2 |
| And/Theo1-NPAM | $+5.8 \times 10^{-3}$ | 0.310 | 0.670 | 5.4% | 19.5 |

The possible effect of solution stirring on $\Delta\Phi_{mbr}$ values associated with concentration polarization was also considered, and Figure 6 shows their dependence on the average concentration value ($C_{avg}$) for Sfw-NPAM ($C_f$ = 0.01 M NaCl) and And-NPAM ($C_f$ = 0.001 M and 0.01 M NaCl) samples, where slight differences depending on both membrane geometrical characteristics and concentration level can be observed; in fact, for this latter membrane practically any effect of concentration polarization in the measures performed at $C_f$ = 0.01 M, are obtained due to the low value of the effective fixed charge concentration.

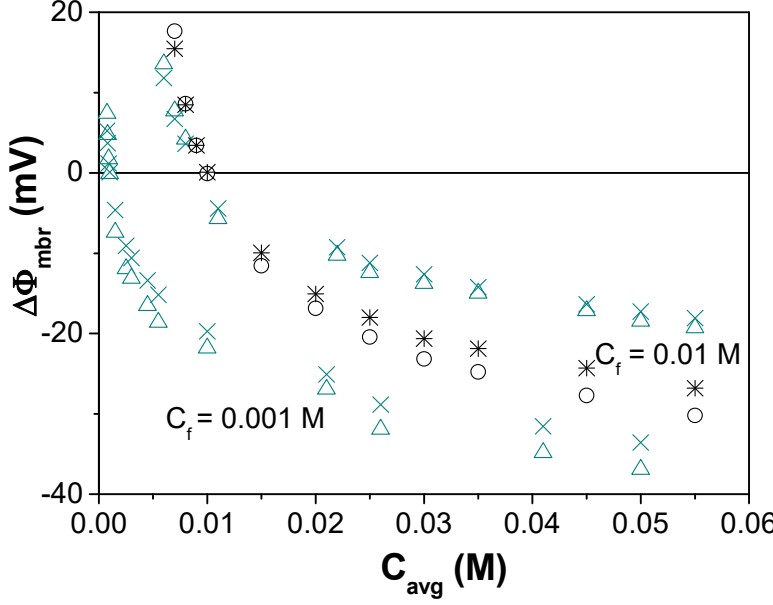

**Figure 6.** Variation of membrane potential with average concentration of the solution at both membranes sides for: Sfw-NPAM (o) stirred solutions and (*) non-stirred solutions, And-NPAM ($\triangle$) stirred solutions and (x) non-stirred solutions.

Figure 7a shows the effect on the electrochemical behavior of the And-NPAM sample of both BSA fouling and surface modification with Theo **1**. According to these results, surface coverage by Theo **1** increases its electropositive behavior since the $\Delta\Phi_{mbr}$ values are closer to those corresponding to an ideal anion-exchanger membrane in the whole range of concentrations; however, the presence of BSA practically eliminates the weak electropositive character of the And-NPAM support without affecting the transport of ions through the membrane according to the similar values obtained at the higher solution concentrations Figure 7b shows theoretical (solid line) and experimental (points) values of membrane potential determined for the And/Theo1-NPAM and the contributions of diffusion potential in the membrane pores (dashed line) and Donnan potential, while the values of electrochemical parameters $X_{ef}$, t+/t− and P(-) obtained by the fit of $\Delta\Phi_{mbr}$ values for And/Theo1-NPAM and And/BSA-NPAM samples are also indicated in Table 2. These results show that the deposition of BSA on the surfaces of the And-NPAM sample

decreases four times the effective fixed charge concentration, while the inclusion of Theo **1** in the original And-NPAM sample increases effective fixed charge concentration and permselectivity (4.5 times and 97%, respectively).

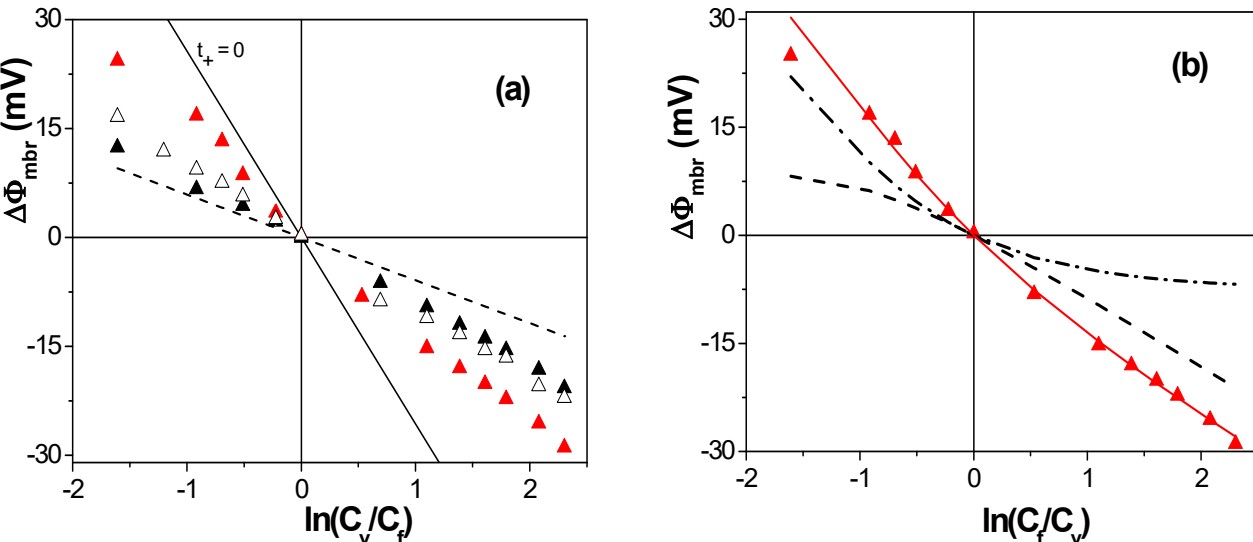

**Figure 7.** (**a**) Comparison of membrane potential dependence with solution concentrations ratio for And-NPAM (△), And/BSA-NPAM (▲) and And/Theo1-NPAM (▲) samples; $\Delta\Phi_{mbr}$ values for an ideal anion exchange membrane (dashed-dot line) and NaCl diffusion potential (dashed line) (**b**) Comparison of experimental (▲) and fitted (solid line) values for And/Theo1-NPAM, Donnan (dashed dot line) and diffusion (dashed line) contributions.

## 4. Conclusions

Analysis by the TMS model of membrane potential measurements provides qualitative and quantitative information on electrochemical parameters affecting the transport of ions through membranes with different characteristics (geometry, structure, or material), but it also allows us to have separate information on interfacial and diffusive contributions. In particular, our results show the electropositive character of all the studied membranes and similar $\Delta\Phi_{mbr} - \ln(C_v/C_f)$ trends independently of membranes characteristics. A significant reduction in the anionic permselectivity with the increase in average pore size was determined for ideal and non-ideal nanoporous membranes, while porosity variation seems to affect diffusion potential contribution (independently of structure and material), only for membranes with pore size > 100 nm, which exhibit rather similar electrical behavior (independently symmetric/asymmetric structure or even surface material). Other factors able to affect ionic transport (concentration polarization or solution concentration level) are also indicated.

Membrane potential analysis has also allowed us the estimation of electrochemical changes in the membranes associated with both protein (BSA) fouling and macromolecule (Theo **1**) inclusion, confirming membrane modification indirectly. Theo **1** inclusion increases the electropositive character of the support membrane, while BSA deposition reduces the effective fixed charge concentration of membranes, but the modification of ions transport values also supports BSA presence on the pore walls, providing valuable information on membrane fouling mechanisms.

**Supplementary Materials:** The following supporting information can be downloaded at: https://www.mdpi.com/article/10.3390/micro2030029/s1, Figure S1: SEM micrographs: (a) supported structure and surface (insert) of CRF25 membrane, and (b) cross section; black bars correspond to 1 μm. (c) top surface of the And-NPAM, black bar is 100 nm; (e) cross section of the And-NPAM, white bar corresponds to 1 μm; (e) down surface of the And-NPAM, black bar corresponds to 500 nm. (f) Sfw-NPAM surface, line corresponds to 1 μm and (g) pore size distribution diagram of the

Sfw-NPAM; Figure S2: Theophylline-oligo(ethylene glycol)-alkene derivative or Theo **1**; Figure S3: Variation of membrane potential values with solution concentration ratio.

**Author Contributions:** Samples selection: J.B. and V.R.: Membrane potential measurements and analysis: V.R. and J.B.; writing and supervision, J.B.; funding acquisition, J.B. All authors have read and agreed to the published version of the manuscript.

**Funding:** This research was funded by Junta de Andalucía (Spain), under grant number FQM 258 (Research Group).

**Conflicts of Interest:** The authors declare no conflict of interest.

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
