# Peer review of "Electrochemical Characterization of Nanoporous Alumina-Based Membranes with Different Structure and Geometrical Parameters by Membrane Potential Analysis"

_2673-8023, doi:10.3390/micro2030029_

Round 1

Reviewer 1 Report

The authors present some interesting data on membrane potential across alumina membranes of different structure and pore size. The data acquired versus concentration difference is analyzed to extract transport numbers, fixed charge concentration in the membrane and anionic permselectivity. Modest but noteworthy variations between the four different membranes studied are found and explained. The effect of protein fouling and biomolecule adsorption is determined to be significant. Some improvements should be made to the presentation and discussion:

(1) line 247 refers to CRF250 and should be CRF25

(2) In figure S1, there are no scale bars or they are too small to see.

(3) Explain as best possible what happens in between the two sides of the asymmetric membrane with 20 nm pores on one side and 200 nm pores on the other side, is it know how gradual or abrupt the change takes place?

(4) The pore structure of CRF membranes which it seems are interparticle gaps should be explained in more detail

(5) In the conclusion, please summarize key differences between each membrane type that was studied

Reviewer 2 Report

Authors experimentally investigate the electrochemical characterization of alumina-based membranes obtained by two different techniques, sinterization or anodization, by membrane potential analysis. The paper presents an interesting qualitative and quantitative information on electrochemical parameters affecting the transport of ions. However, many ambiguous issues should be addressed before the consideration of acceptance of this paper.

1. The introduction focuses on the features of the electrochemical technique without the sintered. Please compare the two fabrication methods in the introduction section.

2. How to obtain Eq. 1 from the formula: ΔФmbr = ∆øDon(I) + ∆ødif + ∆øDon(II), please deduce it specifically. Additionally, the meaning of each variable in Eq. 1 is not elaborated.

3. What is the intuitive annotation of the relationship between the membrane potential and solution concentrations ratio? Please explain this in more detail in the text.

4. Why is the trend of membrane potential dependence with solution concentrations ratio for the ideal charged membrane shown in Fig. 1a? The explanation and discussion should be enhanced.

5. The figures are too monotonous in the text. It is recommended to use symbols of different colors to distinguish the samples, and please use the three-line table in the text.

6. There are several typos and grammatical errors. Please correct them.

7. Please point out the significance of this work and the progress of the current research status in the conclusion section.

Reviewer 3 Report

In the proposed mansucript the Authors described electrochemical characterization of Al2O3-based membranes by analysing values of membrane potential.

The paper is generally well-written and can be publised in Micro after adressing the following issues.

1. Last paragraph of the Introductions section. I do not think that it is necessary to describe the results obtained.

2. Instead of this the elements of novelty of the present work with respect to already published papers should be described.

3. The geometrical features of the particular membranes should be discussed in the main manuscript.

4. What about the reproducibility of the measurements?

5. Can the properties of membrane change after the long exposure to the electrolyte?

6. Do you expect the effect of chemical composition of the membrane? For instance, it is well known that anions from the electrolyte used for anodization can be incorporated to the anodic film, so membranes obtained at different electrolytes can be chemically different.
